# Explicit Conditional Consistency Diffusion: Towards Precise Semantic Alignment in Multimodal Face Generation

## Abstract

With the collaborative guidance of multimodal conditions (e.g., semantic masks as structural visual guidance and text descriptions as linguistic guidance), diffusion models have significantly improved the controllability of face generation. However, existing methods rely solely on noise learning or flow matching to implicitly model the relationship between latent representations and multimodal features, making it difficult to fully capture their semantic associations and resulting in sub-optimal conditional consistency in the generated outputs. To overcome this limitation, we propose $EC^2Face$, a novel facial generation method based on an explicit conditional consistency diffusion framework. Our approach introduces a temporally modulated conditional consistency guidance mechanism in the pixel space, explicitly driving precise semantic alignment between the latent representations and multimodal conditions. In addition, to address the poor response to long-tailed attributes in mask conditions, we design a long-tail adaptive learning strategy. It dynamically assigns differentiated weights to spatial locations through gradient reweighting, enhancing the model's ability to perceive rare attributes and effectively mitigating model bias. Extensive experiments demonstrate that $EC^2Face$ significantly outperforms other competing methods across most evaluation metrics, particularly exhibiting an improvement of over 9.0% in mask accuracy for regions with rare attributes.

## 1 Introduction

Facial generation Ning et al. (2023); Melnik et al. (2024); Wang et al. (2025a), as a core task in the fields of artificial intelligence and computer vision, has long been a focal point of research attention. In recent years, multimodal facial generation has gradually emerged as a research hotspot due to its exceptional controllability advantages. Unlike traditional unimodal generation methods, multimodal facial generation Meng et al. (2025); Sowmya & Meeradevi (2024); Kim et al. (2024b) integrates multi-source modality data, such as textual descriptions (e.g., "blonde hair, blue eyes, smiling expression") and semantic masks (e.g., segmentation masks for facial organ regions), thereby enabling fine-grained control over the generation process .

In the early stages, research on multimodal facial generation was primarily based on Generative Adversarial Networks (GANs) Du et al. (2023a); Meng et al. (2025). However, constrained by factors such as limited expressive capacity in the latent space and unstable training, these methods commonly suffered from issues such as insufficient generation diversity and mode collapse. Diffusion models Po et al. (2024); He et al. (2025); Chang et al. (2025) have achieved a revolutionary breakthrough in the field of image generation, thanks to their unique "forward noising - reverse denoising" progressive learning mechanism. Their flexible conditional injection mechanism Zhang et al. (2023b); Mou et al. (2024a); Peng et al. (2024) has opened up a new technical pathway for the controllability of multimodal facial generation, making them the prevailing technical framework in current research.

Nevertheless, diffusion models primarily rely on noise learning or flow matching Rombach et al. (2022b); Dao et al. (2023) for implicit modeling, making it challenging to fully capture the semantic correlations between latent representations and multimodal conditions. Additionally, the original

optimization mechanism cannot effectively tackle the long-tailed attribute distribution issue in mask conditions, leading to inadequate responses for some rare attributes (e.g., earrings, necklaces).

To address the aforementioned challenges, this paper introduces EC$^2$Face–a novel approach grounded in an explicit conditional consistency diffusion framework. This method significantly enhances the semantic alignment accuracy of synthesized faces through two key improvements. First, we introduce a temporally modulated conditional consistency guidance mechanism in the pixel space. This mechanism explicitly drives the latent representations to maintain precise semantic alignment with multimodal conditions. The specially designed temporal modulation function can circumvent the adverse effects caused by the unreliability of reverse estimation at large time steps. Second, to tackle the modeling bias issue induced by the long-tailed attribute distribution in mask conditions, we propose a long-tailed adaptive learning strategy. This strategy dynamically applies differential weights to spatial positions during the flow matching process Dao et al. (2023) based on pre-computed attribute frequencies, thereby enhancing the sensitivity of the diffusion model to rare attributes and further strengthening the multimodal conditional consistency of synthesized faces.

In summary, our core innovations and contributions are outlined as follows:

1. We propose EC$^2$Face, a novel multimodal facial generation framework with explicit conditional consistency guidance, which addresses the limitations inherent in diffusion models that rely solely on latent space modeling.

2. We devise a multimodal conditional consistency constraint in the pixel space based on latents reverse estimation. This constraint explicitly steers latent representations towards achieving precise semantic alignment with multimodal conditions. Furthermore, we introduce a temporal dynamic modulation function that can effectively circumvent the constraint bias caused by the unreliability of reverse estimation at large time steps.

3. We propose a long-tail adaptive learning strategy, which enhances the model's sensitivity to rare attributes by assigning differential weights to different spatial locations during the flow matching computation process, thereby effectively mitigating the modeling bias of diffusion models towards long-tail semantic attributes.

Comprehensive qualitative and quantitative evaluations demonstrate that EC$^2$Face significantly outperforms existing baseline methods across multiple evaluation metrics, particularly in multimodal conditional consistency.

## 2 RELATED WORKS

### 2.1 FACE GENERATION

Early research on face generation primarily focused on unimodal scenarios, employing only text Pinkney & Li (2022); Park et al. (2023); Zhang et al. (2024) or semantic masks Tan et al. (2021); Wei et al. (2022); Wang et al. (2024) as single input conditions. Among them, representative methods based on Generative Adversarial Networks (GANs), such as the StyleGAN series Karras et al. (2019; 2020) and its text-driven extension StyleCLIP Patashnik et al. (2021), have demonstrated the immense potential of latent space modeling, achieving not only highly realistic generation results but also enabling semantic image editing through latent space manipulations. Despite the significant progress these methods have made in generating realistic and diverse facial images, they are inherently limited by their unimodal nature, exhibiting significant limitations: text-driven approaches often suffer from deviations in spatial layout control, whereas mask-driven methods find it challenging to flexibly manipulate appearance attributes like makeup and accessories. Consequently, unimodal methods face challenges in achieving simultaneous and precise control over both spatial structure and attribute configuration.

To overcome these limitations, multimodal face generation has gradually emerged as a research hotspot, enhancing the controllability of generated faces by integrating the complementary strengths of heterogeneous modalities. For instance, methods such as TediGAN Xia et al. (2021), PixelFace+ Du et al. (2023b), CoDiffusion Huang et al. (2023), and MM2Latent Meng et al. (2025) combine textual descriptions with visual conditions (e.g., masks, sketches), enabling precise multi-dimensional control over identity, pose, and attributes. Similarly, hybrid strategies that fuse diffusion models

with GANs Kim et al. (2024a); Nair et al. (2023) further enhance cross-modal semantic alignment capabilities. However, current methods still face challenges in achieving precise semantic alignment between synthesized faces and multimodal conditions (e.g., masks and text), particularly for modeling long-tailed or small-area attributes such as earrings.

## 2.2 DIFFUSION MODELS

In the field of generative modeling, recent research has demonstrated a paradigm shift from Generative Adversarial Networks (GANs) Salimans et al. (2016); Arjovsky et al. (2017) to diffusion models Ho et al. (2020); Song et al. (2021); Rombach et al. (2022a). This transition is primarily attributed to the significant advantages of diffusion models in terms of training stability and output diversity. Notably, DDPM Ho et al. (2020) and DDIM Song et al. (2021) have laid the technical foundation for diffusion-based generation, while LDM Rombach et al. (2022a) has substantially enhanced image generation efficiency through latent space diffusion strategies. In the realm of controllable generation, the introduction of techniques such as CFG Ho & Salimans (2022), ControlNet Zhang et al. (2023a), T2I-Adapter Mou et al. (2024b), and IP-Adapter Ye et al. (2023) has established diffusion models as the mainstream approach for conditional image generation. Moreover, incorporating visual transformer architectures has allowed us to overcome the perceptual constraints inherent in traditional U-Net models, significantly enhancing the quality of generated images Peebles & Xie (2023); Wang et al. (2025b); Tan et al. (2024); Li et al. (2023). However, existing methods still predominantly rely on noise learning Ho et al. (2020) or flow matching Dao et al. (2023) to implicitly explore the associations between latent variables and multimodal features during multimodal collaborative image generation, resulting in insufficient fine-grained consistency between generated images and multimodal conditions.

To overcome these limitations, we propose a novel multimodal face generation method termed EC$^2$Face. By incorporating a conditional consistency loss function tailored to multimodal conditions in pixel space, our approach effectively compensates for the suboptimal generation quality stemming from implicit latent space modeling. Additionally, we refine the traditional flow matching Dao et al. (2023) learning strategy through differential spatial weight allocation, significantly enhancing the diffusion model's accuracy in modeling long-tailed attributes.

## 3 METHOD

### 3.1 PRELIMINARIES

**Diffusion Transformer.** The Diffusion Transformer (DiT) Peebles & Xie (2023) stands as one of the cutting-edge architectures in the current generation field. It replaces the traditional U-Net backbone with stacked transformer blocks, successfully breaking through the limitations of local perception. The core advantage of this architecture lies in the fact that multimodal feature tokens can engage in deep interactions through the multi-head attention mechanism, with its mathematical formalization expressed as:

$$\text{attn}(q, k, v) = \text{softmax}\left(\frac{qk^\top}{\sqrt{d_h}}\right) v. \tag{1}$$

Here, the query vector $q$, the key vector $k$, and the value vector $v$ are all generated through linear projections of the concatenated multimodal tokens, while $d_h$ denotes the dimensionality of a single attention head.

**Flow matching.** Flow Matching Dao et al. (2023), as an optimization method for generative models based on continuous normalized flows, exhibits significant advantages in generation efficiency and stability compared to the noise prediction paradigm relied upon by traditional diffusion models. Its core lies in constructing a smooth flow field $\{z_t\}_{t\in[0,1]}$ from a noise distribution $\epsilon \sim \mathcal{N}(0,1)$ to a real data distribution $z \sim p_{\text{data}}(z)$, where $t$ serves as a continuous time variable.

The model enhances the generative process by learning the velocity function $v_\theta(z_t, t)$ of the flow field, ensuring that the flow field at any moment $t$ satisfies the following differential equation:

$$\frac{dz_t}{dt} = v_\theta(z_t, t). \tag{2}$$

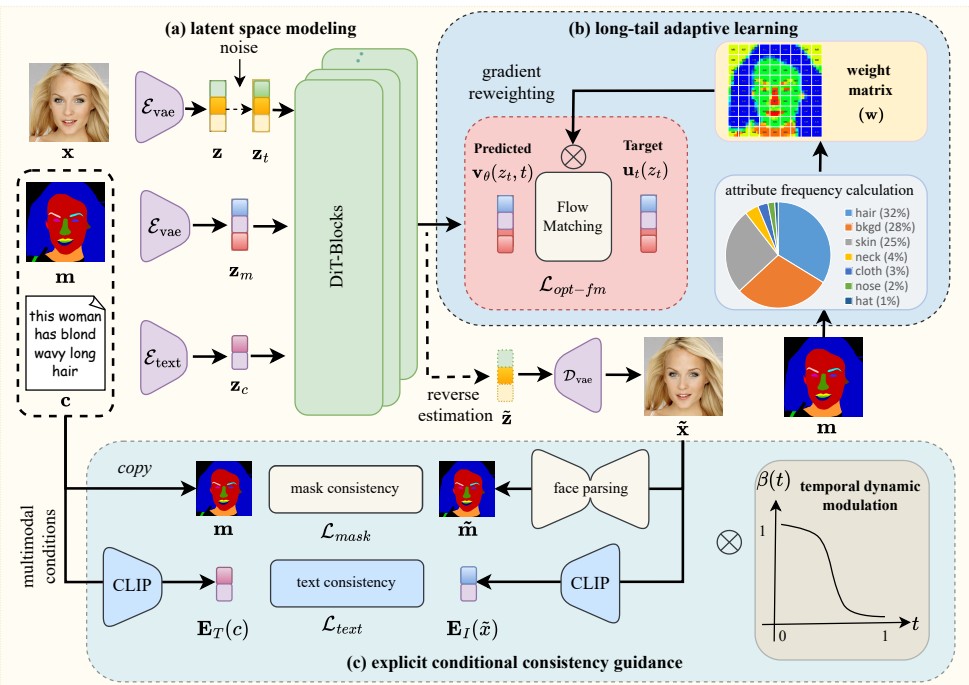

Figure 1: The overall training framework of EC²Face.

The optimization objective of flow matching is to minimize the discrepancy between the predicted velocity field $v_\theta(z_t, t)$ and the target velocity field $\mu_t(z_t)$. The loss function is defined as:

$$\mathcal{L}_{\text{fm}} = \mathbb{E}_{t \sim \text{Uniform}[0,1], \epsilon \sim \mathcal{N}(0,1), z \sim p_{\text{data}}} \left[ \| v_\theta(z_t, t) - \mu_t(z_t) \|_2^2 \right]. \tag{3}$$

Here, $z_t = (1 - t) \cdot z + t \cdot \epsilon$ represents the intermediate state at time $t$ along the flow path between the noise $\epsilon$ and the real sample $z$, and $\mu_t(z_t) = \epsilon - z$ denotes the target velocity field.

## 3.2 OVERALL FRAMEWORK

Figure 1 illustrates the overall training framework of EC²Face. This framework, an extension of FLUX.1-dev Black-Forest-Labs (2024), is designed to precisely synthesize high-fidelity facial images through the collaborative driving of mask and text. During the training process, we project both text and mask information into a latent space shared with noisy images, thereby eliminating discrepancies between representations of heterogeneous modalities. To maintain the framework's simplicity, we reuse the Variational Autoencoder (VAE) to process mask images. Specifically, the noisy image token $z_t$, mask token $z_m$, and text token $z_c$ form a joint token sequence, which is fed into stacked transformer modules for attention-based interaction. The query vector $q$, key vector $k$, and value vector $v$ can be computed according to Equation 4:

$$q = [w_q^c z_c; w_q z_t; w_q^m z_m], \quad k = [w_k^c z_c; w_k z_t; w_k^m z_m], \quad v = [w_v^c z_c; w_v z_t; w_v^m z_m]. \tag{4}$$

In our training framework, we have implemented two core improvement measures to strengthen the semantic consistency between synthetic facial images and multimodal conditions. Firstly, we constructed an explicit conditional consistency guidance mechanism in the pixel space. This mechanism can directly facilitate precise semantic alignment between latent representations and multimodal conditions. Secondly, to address the modeling bias issue of diffusion models when handling long-tailed attributes, we innovatively proposed a long-tail adaptive learning strategy. This strategy effectively enhances the model's ability to model rare attributes, thereby significantly improving the controllable accuracy of synthetic facial images.

### 3.3 EXPLICIT CONDITIONAL CONSISTENCY GUIDANCE

Existing DiTs models predominantly employ flow-matching learning strategies within the latent space. However, such implicit modeling approaches struggle to enable latent representations to accurately capture the semantic information of multimodal features (e.g., text $c$ and mask $m$). To address this challenge, this study proposes introducing explicit constraint functions compatible with multimodal conditions in the pixel space to directly drive the modeling process of diffusion transformers. Specifically, during the training phase, for each noisy latent $z_t$ and the model-predicted velocity field $v_\theta(\mathbf{z}_t, t)$, the initial latent variable $\tilde{z}$ is derived through reverse estimation (as shown in Equation 5), and then an estimated facial image $\tilde{x}$ is generated via the VAE decoder. Based on $\tilde{x}$, the consistency loss between the facial image and multimodal conditions can be quantitatively computed in the pixel space.

$$\tilde{z} = z_t - t \cdot v_\theta(z_t, t). \tag{5}$$

**Text Consistency Loss.** We employ the CLIP model Radford et al. (2021) to measure the semantic consistency between the facial image $\tilde{x}$ and the textual condition $c$. This model is equipped with a pre-trained text encoder $\mathbf{E}_T$ and a pre-trained image encoder $\mathbf{E}_I$, which can efficiently extract features from both text and images. Based on the extracted features, we utilize the cosine loss function to quantify the discrepancy between the generated facial image $\tilde{x}$ and the given textual condition $c$. The specific formula is as follows:

$$\mathcal{L}_{\text{text}} = 1 - \frac{\mathbf{E}_I(\tilde{x}) \cdot \mathbf{E}_T(c)}{\|\mathbf{E}_I(\tilde{x})\| \cdot \|\mathbf{E}_T(c)\|}. \tag{6}$$

**Mask Consistency Loss.** We employ the pre-trained facial parsing model FaRL Zheng et al. (2022) to process the facial image $\tilde{x}$, thereby obtaining its predicted semantic segmentation map $\tilde{m}$. Based on this predicted outputs and the given mask condition $m$, we calculate the mask consistency loss by taking a weighted combination of the multi-label cross-entropy loss and the Dice coefficient loss Milletari et al. (2016). The specific definition can be found in Equation 7.

$$\mathcal{L}_{\text{mask}} = \lambda \cdot \mathcal{L}_{\text{CE}}(\tilde{m}, m) + (1 - \lambda) \cdot \mathcal{L}_{\text{Dice}}(\tilde{m}, m)$$

$$= \lambda \cdot \left[ -\sum_i \left( m_i \log \tilde{m}_i + (1 - m_i) \log(1 - \tilde{m}_i) \right) \right]$$

$$+ (1 - \lambda) \cdot \left( 1 - \frac{2 \sum_i m_i \tilde{m}_i}{\sum_i m_i^2 + \sum_i \tilde{m}_i^2 + \delta} \right), \tag{7}$$

where $\lambda \in [0, 1]$ is the balancing coefficient (set to 0.5 empirically in our experiments), and $\delta > 0$ denotes a smoothing parameter introduced to ensure numerical stability.

**Temporal Dynamic Modulation.** The acquisition of facial image $\tilde{x}$ hinges on the latent reverse estimation outlined in Equation 5 (from $z_t$ to $\tilde{z}$). Experiments reveal that the reliability of this reverse estimation is not uniform across time steps; as shown in Figure 2, it declines as the time step $t$ increases. To avoid the adverse effects caused by the unreliability of reverse estimation at large time steps, we designed a temporal dynamic modulation function, the specific form of which is presented in Equation 8. This function can dynamically adjust the intensity of the conditional consistency constraint according to the variation of the timestep $t$.

$$\beta(t) = \frac{1}{1 + \exp\left(\gamma_s \cdot (t - t_c)\right)}. \tag{8}$$

The rationale is straightforward: When $t$ is small, the reverse estimation is highly reliable, so we should strengthen its influence. Conversely, when $t$ is large, the reverse estimation becomes less reliable, necessitating a reduction in the constraint intensity. The coefficients $\gamma_s$ and $t_c$ control the steepness of this function and the position of its center point, respectively. We empirically set $\gamma_s = 10$ and $t_c = 0.4$ in our experiments.

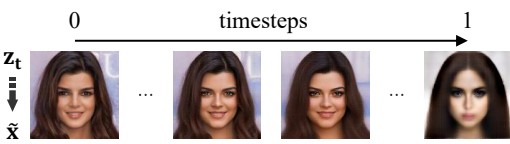

Figure 2: Reliability of reverse estimation.

**Multimodal Conditional Consistency Loss.** Upon applying temporal dynamic modulation, we can calculate an overall multimodal conditional consistency loss in the pixel space, with its specific formulation depicted in Equation 9. Here, $\alpha_1$ and $\alpha_2$ serve as balancing coefficients to adjust the contribution weights of different loss terms.

$$\mathcal{L}_{\text{mcc}} = \beta(t) \cdot (\alpha_1 \cdot \mathcal{L}_{\text{text}} + \alpha_2 \cdot \mathcal{L}_{\text{mask}}).\tag{9}$$

### 3.4 Long-tail Adaptive Learning

**Long-Tail Distribution of Semantic Attributes.** We observed a significant imbalance in the distribution of semantic attributes within mask images (refer to Figure 5 in Appendix A for details). Statistical analysis of the widely - used MM-CelebA-HQ dataset Lee et al. (2020) revealed that semantic attributes exhibit a typical long-tailed distribution in terms of both average pixel proportion and frequency of occurrence in training samples. However, existing methods fail to effectively address this issue, resulting in rare attributes such as earrings and necklaces being overlooked during the generation process. To tackle this challenge, we propose a long- ailed adaptive learning strategy that enhances the diffusion model's perception of rare attributes through a gradient reweighting mechanism.

**Gradient Reweighting.** In the flow-matching computation within the latent space, we employ a method of assigning differential weights to spatial positions to enhance the model's perception of rare attributes. Specifically, during the data loading phase, we pre-compute the distribution of semantic attributes within the mask $m$. Let $n_c$ denote the total number of attribute categories, and $\{T_c\}_{c=1}^{n_c}$ represent the aggregated pixel count for each attribute category. Then, the elements of the differential weight matrix $w \in \mathbb{R}^{h \times w}$ are calculated according to the following formula:

$$w(i,j) = \frac{\sum_{c=1}^{n} T_c}{n_c \cdot T_{m(i,j)} + 1}\tag{10}$$

where $m(i,j)$ indicates the attribute category label at the spatial position $(i,j)$. The underlying motivation of this approach is to increase the weights of regions with rare attributes and decrease the weights of regions with high-frequency attributes, thereby achieving a relative balance among different attributes. The calculation of the flow-matching loss after gradient reweighting is shown in Equation 11:

$$\mathcal{L}_{\text{opt-fm}} = \mathbb{E}_{t \sim \text{Uniform}[0,1], \epsilon \sim \mathcal{N}(0,1), z \sim p_{\text{data}}} \left[ \| w \cdot (v_\theta(z_t, m, c, t) - \mu_{\mathbf{t}}(z_t)) \|_2^2 \right].\tag{11}$$

### 3.5 Training strategy

**LoRA Fine-tuning.** During the training phase, to reduce computational overhead and parameter redundancy in model training, we employ a lightweight adaptation architecture. Specifically, we conduct low-rank adaptation fine-tuning (LoRA) Hu et al. (2021) solely on the visual embedding layer of the base model and some linear layers within the transformer blocks. Compared with full-model fine-tuning, this approach introduces an increase of less than 0.1% in the number of new parameters. The final loss function of this model comprises two components: firstly, the flow-matching loss with gradient reweighting; secondly, the multimodal conditional consistency loss introduced in the pixel space:

$$\begin{aligned}\mathcal{L}_{\text{total}} &= \mathcal{L}_{\text{opt-fm}} + \mathcal{L}_{\text{mcc}} \\ &= \mathcal{L}_{\text{opt-fm}} + \beta(t) \cdot (\alpha_1 \cdot \mathcal{L}_{\text{text}} + \alpha_2 \cdot \mathcal{L}_{\text{mask}}).\end{aligned}\tag{12}$$

**Stochastic Condition Dropout.** To endow the model with the ability to handle both multimodal collaborative-driven and unimodal-driven facial generation simultaneously, we specifically introduce a stochastic condition dropout strategy during the training process. More precisely, at the data loading stage, for each individual sample, this strategy independently and randomly drops the text condition $c$ and the mask condition $m$ with a pre-defined probability p ($p = 0.1$ in our experiments). The detailed procedure is demonstrated in Equation 13.

$$c/m = \begin{cases} \phi & \text{with probability } p, \\ c/m & \text{with probability } 1-p. \end{cases}\tag{13}$$

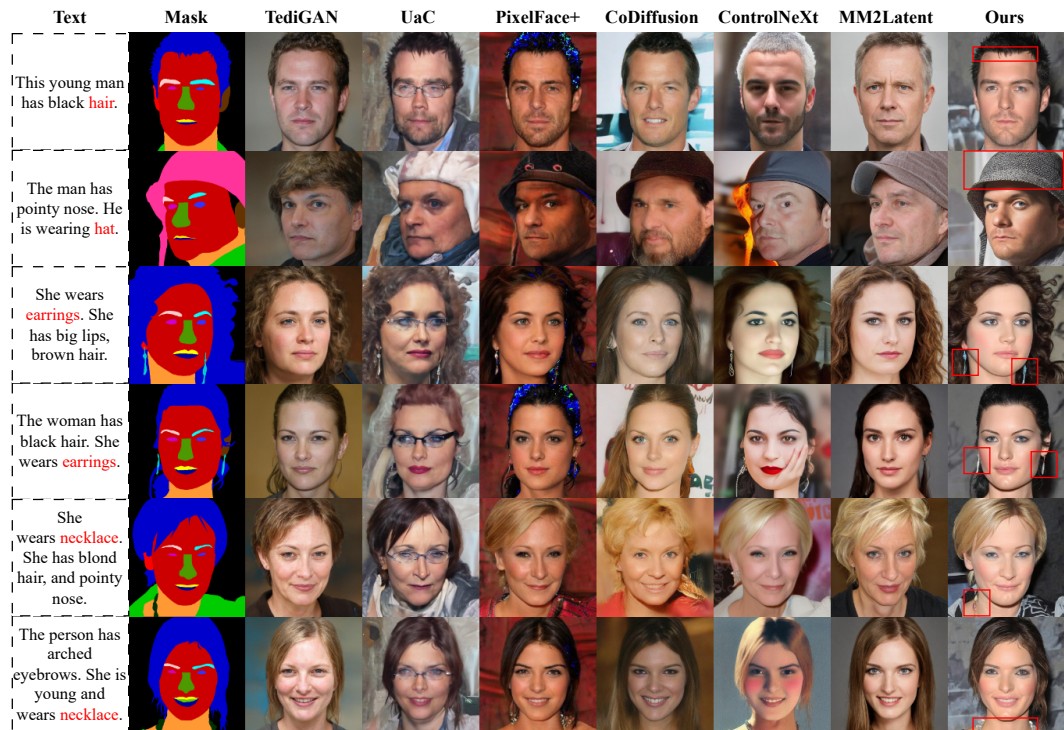

Figure 3: Visual comparison of multimodal face generation.

# 4 EXPERIMENTS

## 4.1 EXPERIMENTAL SETUP

**Datasets.** To comprehensively evaluate the performance of the proposed EC$^2$Face model, we selected the MM-CelebA-HQ Lee et al. (2020), a large-scale benchmark dataset widely recognized in the field of multimodal facial synthesis. This dataset comprises 30,000 high-resolution, diverse RGB facial images, each accompanied by 10 distinct textual descriptions and professionally annotated semantic masks. These masks precisely partition the face into 19 attribute categories, encompassing both facial features (e.g., skin, eyes, mouth) and accessories (e.g., earrings, necklaces). In our experiments, the MM-CelebA-HQ dataset was divided into training and test sets at a ratio of 9:1, which were respectively utilized for model training and performance evaluation. In addition, we employed a pre-trained facial parser Zheng et al. (2022) to process the FFHQ-Text dataset Zhou (2021), obtaining facial masks. Furthermore, we process the FFHQ-Text dataset using a pre-trained face parser to obtain facial masks, based on which we construct the MM-FFHQ dataset. This new benchmark allows us to further evaluate the zero-shot generalization capability of EC$^2$Face.

**Evaluation Metrics.** We employ widely recognized evaluation metrics to measure the performance of our model. Specifically, we utilize the CLIP score Radford et al. (2021) to assess the consistency between facial images and given textual conditions, while leveraging the accuracy metric to calculate the average accuracy of various attributes in facial images in mask conditions. Regarding image quality assessment, we adopt the Learned Perceptual Image Patch Similarity (LPIPS) Zhang et al. (2018) metric to quantify the perceptual similarity between generated images and real images, and employ the Natural Image Quality Evaluator (NIQE) Mittal et al. (2012) to evaluate the naturalness quality of the synthesized facial images.

**Implementation Details.** All experiments were conducted on a server equipped with eight NVIDIA A100 GPUs (each with 80GB memory). We employed the AdamW optimizer with an initial learning rate of $1 \times 10^{-4}$. During training, each GPU processed a mini-batch of four samples, achieving an effective global batch size of 32 through multi-GPU parallel computation. The model underwent 5,000 optimization iterations with gradient accumulation enabled to ensure stable training conver-

gence. The hyperparameters in the loss function were set as $\alpha_1 = 0.5$ and $\alpha_2 = 0.5$. For reproducibility, we fixed the random seed to 42 across all experiments during inference. The sampling process utilized a Flow-Matching Euler Discrete scheduler Esser et al. (2024) with 28 discretization steps.

## 4.2 QUANTITATIVE COMPARISON

To thoroughly analyze the disparities in generation accuracy across different attribute regions, during the evaluation of mask-conditional consistency, we not only report the mean accuracy for overall attributes (OAA) but also separately calculate the mean accuracies for high-frequency attributes (HFA), normal-frequency attributes (NFA), and rare-frequency attributes (RFA) (refer to Appendix A for detailed attribute categorization).

Table 1 presents the quantitative comparison between the proposed EC$^2$Face method and state-of-the-art approaches on the MM-CelebA-HQ benchmark dataset. The experiments demonstrate that our method significantly outperforms the competing methods in terms of both text and mask conditional consistency. Notably, in terms of the generation accuracy of rare attributes, our method nearly doubles that of the second-best method. Meanwhile, it also achieves state-of-the-arts performance in generated image quality. Furthermore, Table 2 showcases the zero-shot generalization experimental results on the MM-FFHQ dataset, further validating the superiority and robustness of the EC$^2$Face method.

Table 1: Quantitative comparison with state-of-the-art methods on the MM-CelebA-HQ dataset.

| Methods | Multimodal Conditional Consistency | | | | | Visual Quality | |
| | Text (%, ↑) | Mask (%, ↑) | | | | LPIPS (↓) | NIQE (↓) |
| | | OAA | HFA | NFA | RFA | | |
| TediGAN Xia et al. (2021) | 23.90 | 66.58 | 91.14 | 69.18 | 0.61 | **0.49** | 5.19 |
| UaC Nair et al. (2023) | 25.52 | 65.37 | 79.54 | 68.27 | 18.19 | 0.57 | 6.47 |
| PixelFace+ Du et al. (2023b) | 26.16 | 82.11 | 93.96 | 86.80 | 27.81 | 0.56 | 6.24 |
| CoDiffusion Huang et al. (2023) | 24.51 | 61.51 | 85.34 | 62.83 | 5.26 | 0.58 | 4.46 |
| ControlNeXt Peng et al. (2024) | 25.88 | 80.93 | 91.67 | 84.94 | 33.40 | 0.54 | 4.54 |
| MM2Latent Meng et al. (2025) | 24.61 | 65.62 | 85.75 | 69.18 | 2.27 | 0.51 | 3.74 |
| **EC$^2$Face (Ours)** | **27.05** | **88.64** | **96.56** | **90.18** | 62.78 | 0.51 | **3.55** |

Table 2: Comparison of zero-shot generalization performance on the MM-FFHQ dataset.

| Methods | Multimodal Conditional Consistency | | | | | Visual Quality | |
| | Text (%, ↑) | Mask (%, ↑) | | | | LPIPS (↓) | NIQE (↓) |
| | | OAA | HFA | NFA | RFA | | |
| TediGAN Xia et al. (2021) | 25.03 | 64.63 | 91.40 | 66.25 | 0.56 | 0.64 | 5.28 |
| UaC Nair et al. (2023) | 26.97 | 67.61 | 82.52 | 70.23 | 20.73 | 0.67 | 6.39 |
| PixelFace+ Du et al. (2023b) | 26.60 | 83.36 | 93.72 | 88.24 | 30.99 | 0.67 | 5.93 |
| CoDiffusion Huang et al. (2023) | 23.03 | 58.35 | 82.80 | 59.31 | 3.22 | 0.68 | 6.79 |
| ControlNeXt Peng et al. (2024) | 27.80 | 71.63 | 86.93 | 74.72 | 20.95 | 0.62 | 3.76 |
| MM2Latent Meng et al. (2025) | 26.37 | 63.25 | 84.97 | 66.00 | 1.93 | 0.64 | 3.83 |
| **EC$^2$Face (Ours)** | **28.40** | **88.28** | **96.98** | **90.02** | 59.63 | **0.51** | **3.67** |

## 4.3 QUALITATIVE COMPARISON

Figure 3 presents a visual comparison of facial images synthesized by EC$^2$Face and current state-of-the-art methods under the collaborative guidance of mask and text conditions. As clearly depicted

in the figure, the facial images generated by EC$^2$Face achieve more precise semantic alignment between the given text and mask conditions, as exemplified by the hair-end details in the first row, the hat shape in the second row, the earring styles in the third and fourth rows, and the necklace features in the fifth and sixth rows. This indicates that our method demonstrates significant advantages in fine-grained control, such as response to rare attributes, while ensuring high visual fidelity. For more visualization examples, please refer to the Appendix.

## 4.4 ABLATION STUDIES

**Effectiveness of Long-tail Adaptive Learning.** We conducted ablation experiments to evaluate the effectiveness of the long-tail adaptive learning strategy (LTAL). The quantitative results in Table 3 demonstrate that this strategy significantly enhances the mask alignment accuracy for rare attribute regions, with the average accuracy increasing from 52.04% to 61.1%. Further analysis reveals that the strategy yields minimal improvement for high-frequency attributes, while boosting the average accuracy of normal-frequency attributes by 0.79%. Figure 4 provides a visual comparison that intuitively illustrates the improvement in accuracy for rare-frequency attributes.

**Effectiveness of Explicit Conditional Consistency Guidance.** Table 4 presents the results of item-by-item ablation experiments conducted on explicit conditional consistency guidance (ECCG). The experiments demonstrate that applying either the text consistency constraint $\mathcal{L}_{\text{text}}$ or the mask consistency constraint $\mathcal{L}_{\text{mask}}$ individually can enhance the model's evaluation scores under corresponding conditions. Furthermore, when both constraints are concurrently applied, all metrics pertaining to multimodal conditional consistency demonstrate notable improvements, thereby providing conclusive evidence for the effectiveness of the explicit consistency guidance approach.

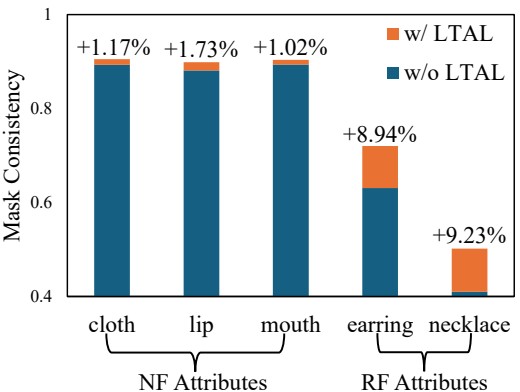

Figure 4: Accuracy enhancement of rare attributes through long-tail adaptive learning.

Table 3: Ablation experiment on the effectiveness of the proposed long-tail adaptive learning.

| LTAL | Mask (%, ↑) | | | |
|---|---|---|---|---|
| | OAA | HFA | NFA | RFA |
| ✗ | 86.14 | 95.13 | 88.62 | 52.04 |
| ✓ | **87.64** | **95.18** | **89.41** | **61.12** (**+9.08**) |

Table 4: Ablation experiment on the effectiveness of explicit conditional consistency constraints.

| ECCG | | Text (%, ↑) | Mask (%, ↑) | | | |
|---|---|---|---|---|---|---|
| $\mathcal{L}_{\text{text}}$ | $\mathcal{L}_{\text{mask}}$ | | OAA | HFA | NFA | RFA |
| ✗ | ✗ | 26.12 | 87.64 | 95.18 | 89.41 | 61.62 |
| ✗ | ✓ | 26.26 | 88.59 | 96.50 | 90.14 | 62.68 |
| ✓ | ✗ | 27.22 | 87.48 | 95.36 | 89.16 | 60.60 |
| ✓ | ✓ | 27.05 | 88.64 | 96.56 | 90.18 | 62.78 |

## 5 CONCLUSION

This study proposes a novel facial synthesis method named EC$^2$Face, grounded on an explicit-consistency-guided diffusion transformer architecture, which enables mask-text multimodal collaborative facial generation. Relying on latents reverse estimation, we introduce a multimodal condition consistency guidance mechanism in the pixel space. This mechanism explicitly drives the latent representations to achieve precise semantic alignment with the multimodal conditions, effectively overcoming the suboptimal generation issues caused by relying solely on implicit modeling in the latent space. Meanwhile, we design a temporal modulation function to avoid constraint biases resulting from the unreliability of reverse estimation at large time steps. Additionally, we propose a long-tail adaptive learning strategy. This strategy, by gradient reweighting during flow matching, mitigates the modeling bias of diffusion models towards rare attributes, leading to an improvement of over 9.0% in average accuracy.

ETHICAL STATEMENT

The multimodal face generation method proposed in this study is capable of high-fidelity facial image generation based on natural language descriptions and semantic masks, with anticipated applications in areas such as virtual character creation and secure identity verification. The positive motivation of this research lies in providing technological support for public safety and social well-being. However, we are fully aware that if this technology is misused, it could be employed to forge or disseminate fake facial images. Therefore, we strongly advocate for the implementation of rigorous human verification and professional supervision mechanisms, and emphasize that the model's output results should not be regarded as conclusive evidence. We encourage the research community to apply this technology in a responsible manner, ensuring its consistent compliance with ethical and legal standards.

REPRODUCIBILITY STATEMENT

We have included the complete source code for model implementation, training, and testing in the supplementary materials. We sincerely invite other researchers to replicate our experimental results based on this code and further expand our work, thereby jointly advancing technological development in the field of multimodal face generation.

THE USE OF LARGE LANGUAGE MODELS

In the course of this research and in preparing the manuscript, we utilized Large Language Models (LLMs) DeepSeek-AI (2024) in a limited capacity. Specifically, during the manuscript preparation phase, an LLM was used to assist in refining the wording and improving the clarity of the English prose. Its role in this capacity was strictly limited to enhancing sentence structure, grammar, and the overall flow of the text. Beyond this, LLMs were not involved in the research design, data collection, evaluation, or the generation of core scientific ideas. All substantive content, methodologies, and conclusions are entirely the original work of the authors.

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

## A  PRE-STATISTICAL ANALYSIS AND SUBSET PARTITIONING OF SEMANTIC ATTRIBUTES IN MASK CONDITIONS.

The mask images in the MM-CelebA-HQ dataset Lee et al. (2020) predefine 19 typical facial semantic attributes. Figure 5 presents statistical analyses of the average pixel proportion of each attribute within the images and their occurrence frequencies across the dataset, revealing a pronounced long-tailed distribution of semantic attributes—manifesting as severe imbalances in both pixel proportion and occurrence frequency. However, existing methods generally lack optimization strategies tailored to this distributional characteristic, resulting in suboptimal generation performance for rare-frequency attributes.

To systematically evaluate the generation quality across different attribute regions, this study categorizes the 19 semantic attributes into three groups based on the statistical results: High-Frequency Attributes (HFA), Normal-Frequency Attributes (NFA), and Rare-Frequency Attributes (RFA). The classification criteria are as follows: an attribute is assigned to the HFA group if it exhibits both high average pixel proportion and high occurrence frequency; to the RFA group if it demonstrates low values in both metrics; and the remaining attributes are classified as NFA. The specific compositions of the three attribute groups are detailed as follows:

- **HFA** (High-Frequency Attributes): `background`, `hair`, `skin`, `neck`.
- **NFA** (Normal-Frequency Attributes): `cloth`, `nose`, `l_lip`, `u_lip`, `l_ear`, `r_ear`, `l_brow`, `r_brow`, `mouth`, `l_eye`, `r_eye`, `hat`, `glasses`.
- **RFA** (Rare-Frequency Attributes): `earring`, `necklace`.

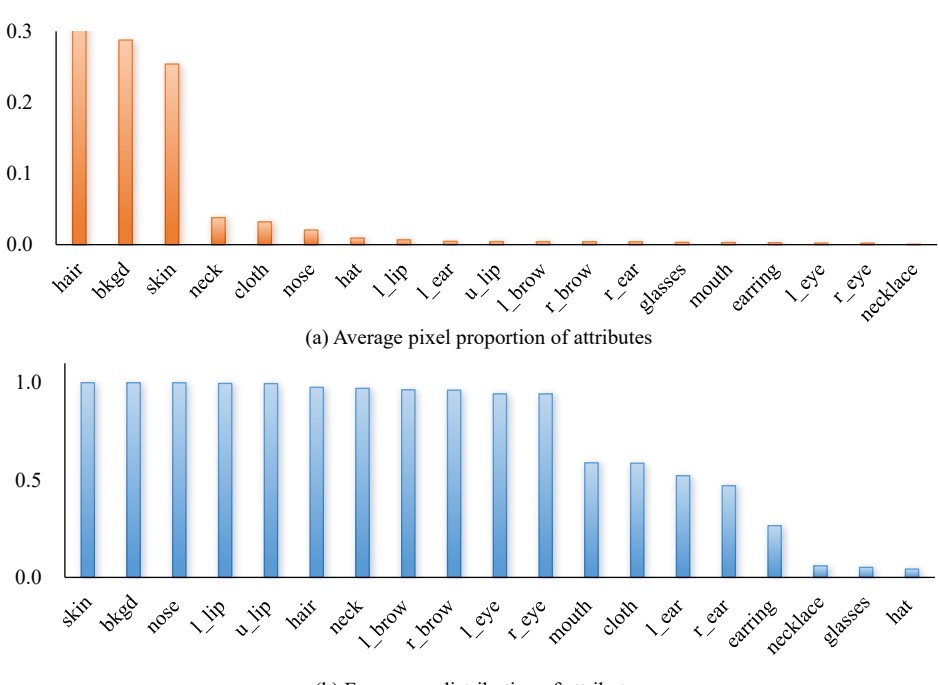

(a) Average pixel proportion of attributes

(b) Frequency distribution of attributes

Figure 5: The long-tailed distribution of semantic attributes in mask conditions.

## B  MORE EXAMPLES OF MULTIMODAL FACIAL GENERATION.

Figure 6 presents more cases of multimodal facial synthesis. On the left side of each row are the given text and mask conditions, while on the right side are the generated results of TediGAN Xia et al. (2021), UaC Nair et al. (2023), PixelFace+ Du et al. (2023b), CoDiffusion Huang et al. (2023), ControlNeXt Peng et al. (2024), MM2Latent Meng et al. (2025), and our proposed EC²Face in

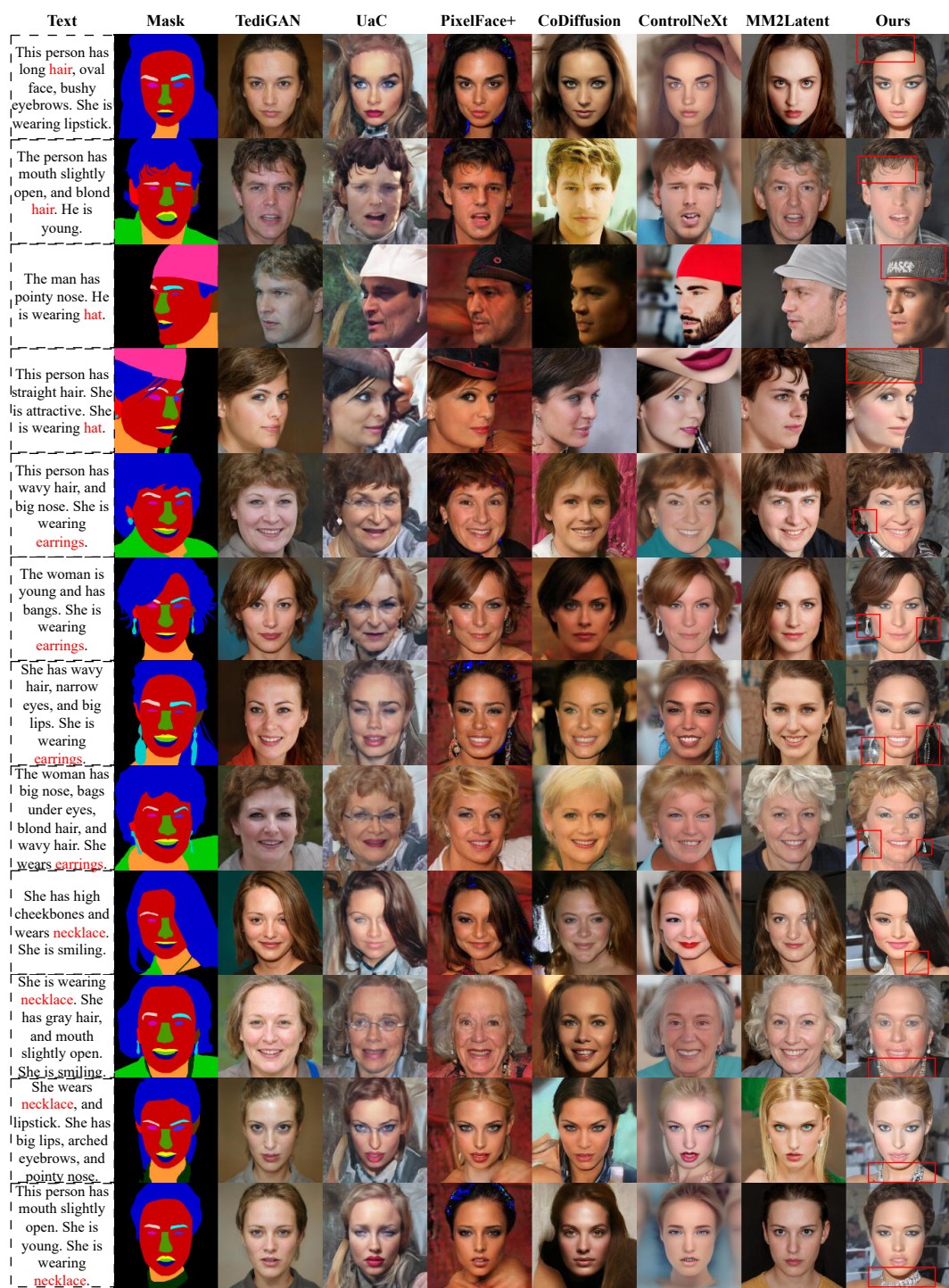

Figure 6: Visual comparison of additional multimodal facial synthesis samples.

sequence. Compared with other competing methods, EC$^2$Face exhibits superior performance in detail control, with the areas of significant improvement marked by red boxes in the figure. This indicates that, given the text and mask conditions, EC$^2$Face can generate facial images that are more semantically aligned with the two modalities (text and image mask) and can more clearly respond to the conditional signals of rare attributes (e.g., earrings, necklaces).

## C    COMPATIBILITY OF UNIMODAL-DRIVEN FACIAL GENERATION

EC$^2$Face simultaneously supports facial generation tasks under both multimodal collaborative and unimodal (text or mask) conditions, without requiring any modification to the model architecture. This capability stems from two core design principles:

1. Explicit conditional consistency guidance in pixel space: By strengthening semantic alignment between latent representations and input conditions, this mechanism ensures generated results strictly adhere to the specified controllable conditions, regardless of whether they are unimodal or multimodal.

2. Stochastic conditional dropout during training: By simulating unimodal input scenarios, this strategy equips the model with robustness against missing modalities, enabling flexible adaptation to diverse conditional combinations during generation.

Experimental validation, as illustrated in Figure 7, demonstrates that EC$^2$Face maintains high-quality facial generation even when provided solely with textual descriptions or mask images. By varying random seeds, the model preserves output diversity while ensuring both semantic consistency and visual fidelity.

## D    GENERATIVE DIVERSITY UNDER CONDITIONAL CONSISTENCY.

We conducted an in-depth exploration of the multimodal generation diversity of EC$^2$Face. In the experiments, with a fixed set of text and mask conditions, we employed different random seeds to drive the model to generate facial images, and some representative samples are shown in Figure 8. The experimental results demonstrate that while ensuring precise semantic alignment between the generated results and the given conditions, EC$^2$Face also exhibits remarkable diversity in unspecified attribute dimensions, covering aspects such as identity traits, hair color, skin tone, earring styles, and hat appearances. This characteristic offers broad application prospects in practical scenarios like data augmentation and interactive creation.

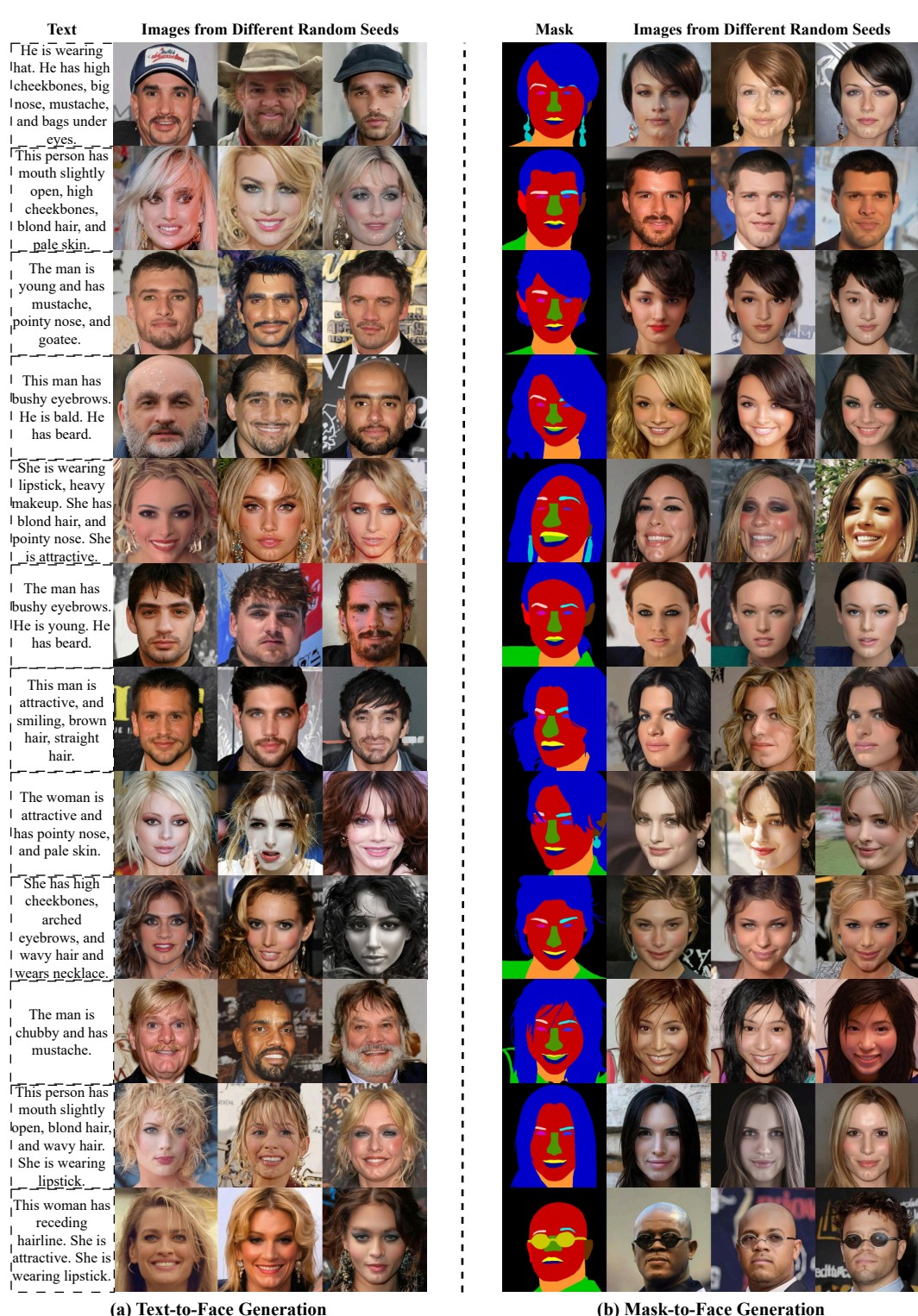

Figure 7: Compatibility of unimodal-driven face generation using EC$^2$Face.

Text Prompt: She has bangs and mouth slightly open.

Text Prompt: This person has big nose, high cheekbones and she is wearing earrings.

Text Prompt: She wears earrings and has bangs, wavy hair, arched eyebrows, pointy nose, black hair.

Text Prompt: He is bald and has oval face.

Text Prompt: This person is young and has mouth slightly open, bags under eyes, and brown hair.

Text Prompt: He is wearing earrings. He has double chin.

Text Prompt: The young woman wears hat and has straight hair.

Text Prompt: This person has sideburns, and straight hair.

Figure 8: Facial generation diversity under multimodal consistency.

