# OpenReview forum: "Explicit Conditional Consistency Diffusion: Towards Precise Semantic Alignment in Multimodal Face Generation"
_ICLR.cc/2026/Conference — ICLR 2026 Conference Withdrawn Submission_

### Official Review · Reviewer_E5bF · 2025-10-30

**Soundness:** 2
**Presentation:** 2
**Contribution:** 1
**Rating:** 2
**Confidence:** 3

**Summary:**

The paper introduces EC2Face, is a facial generation approach built on explicitly conditioning an image diffusion model on face semantic maps. It introduces a temporally modulated conditional consistency guidance mechanism in pixel space to explicitly align latent representations with multimodal conditions such as semantic masks and text descriptions in the pixel space. Additionally, it incorporates a long-tail adaptive learning strategy that reweights gradients across spatial locations to better handle rare attributes in mask conditions and reduce model bias.

**Strengths:**

- The t=0 estimated generation is calculated via eq.5 which linearly extrapolates the current diffusion step, the limitations of the approach are discussed and mitigation for its limitation o=is introduced.

**Weaknesses:**

- The generation domain seems quite limited to centered face portraits
- Furthermore, the useability of the model is limited. Providing explicit semantic maps for faces is a very strong condition that makes inference quite cumbersome in practice as semantic maps are difficult to generate and edit.
- Text and mask consistency both operate on pixel space, making the training algorithm inefficient since it requires passing the generated image (estimated) through x2 models keeping all gradients (please correct if misunderstood). This requires the use of LoRA training, even with a x8 A100 server, which is a major limitation for scaling the method
- The method is not novel, text and image consistency in the form presented here are direct applications which have been explored extensively. There are no tricks to make training more efficient.

**Questions:**

- Which version of FaRL was used for deriving semantic masks?
- Is the model limited to generating CelebAMask-HQ style - limited to frontal view, closeup - face images?

---

### Official Review · Reviewer_iAP4 · 2025-10-31

**Soundness:** 2
**Presentation:** 2
**Contribution:** 2
**Rating:** 4
**Confidence:** 3

**Summary:**

This paper presents a new method for multimodal conditional face generation.

Authors argue existing works only model the relationship between multimodal conditions and the generated face images implicitly, making it difficult to capture their semantic correlations. Plus, existing works are limited to handle the long-tailed attribute distribution issue in mask conditions.

Specific to these problems, this paper presents two contributions:

1. Explicit conditional consistency guidance in pixel space. Authors add explicit mask consistency loss and CLIP-based text alignment loss to help the model align the latent representation with the multimodal consistency. Considering the generated image reversed from noisy latent can be unreliable, authors also present a timestep-dependent strategy to weight the losses adaptively.

2. Long-tailed adaptive learning strategy. To encourage the model pay more attention to rare attributes, authors use the ratio of different category in the mask as the weighting. The flow-matching loss is reweighted by this weighting spatially, emphasizing the rare attribites.

Qualitative and quantitative results on MM-CelebA-HQ and MM-FFHQ benchmark indicate the effectiveness of the proposed method.

**Strengths:**

1. The paper is well motivated and well structured.
2. The proposed method achieves strong performance compared to existing methods.

**Weaknesses:**

1. The strong performance seems to come from the strong base Flux model. According to the ablation study, even without the proposed algorithms, the model can achieve strong results compared to other methods. The benefit brought by the Explicit conditional consistency guidance looks limited.

2. It is well known at the early stage of denoising, the latent is noisy and can produce unreliable estimated $z_0$. The proposed Temporal Dynamic Modulation dynamically assign loss weight according to the time steps. In other words, it compromises the gradients at the early stage by assigning a smaller weight, making it fail to fully utilize the supervision signal. In contrast, recent diffusion-based works like PuLID[1] presents a way to propagate the gradients through the whole denoising trajectory and they claim to utilize the supervision better. I would encourage the authors to provide a discussion about this design.

3. The evaluation looks limited. During evaluation, the input shares the similar distribution with the training data. In qualitative results, most of information in the text prompts are covered by the human mask. I am curious about how it performs for OOD text prompts with diversity in age, nationality and emotions.

[1] PuLID: Pure and Lightning ID Customization via Contrastive Alignment. NeurIPS 2024

**Questions:**

Please see the weakness.

**Details Of Ethics Concerns:**

According to the visual results in the paper, the model seems to associate some text prompts with specific group of people. This may raise bias concerns for human subject image generation.

---

### Official Review · Reviewer_nfq3 · 2025-11-01

**Soundness:** 2
**Presentation:** 2
**Contribution:** 2
**Rating:** 4
**Confidence:** 4

**Summary:**

The paper presents EC2Face, a multimodal face generation method based on the diffusion transformer. It introduces three claimed contributions: Explicit Conditional Consistency Guidance,  Temporal Dynamic Modulation, and Long-Tail Adaptive Learning.
Experiments on MM-CelebA-HQ and MM-FFHQ report moderate gains in CLIP score and mask accuracy.

**Strengths:**

1. Clear paper organization and extensive experimental evaluation.
2. Qualitative results demonstrate reasonable visual improvements.

**Weaknesses:**

1. The proposed Explicit Text Consistency Loss is not new — a very similar idea has already been used in DiffusionCLIP and subsequent text-conditioned editing methods that employ CLIP-based alignment within diffusion frameworks. The Temporal Dynamic Modulation (TDM) is just a scheduling trick, where a logistic weighting is applied over timesteps; such timestep-dependent weighting functions are widely used in diffusion and flow-matching literature. The Long-Tail Adaptive Learning (LTAL) is a basic static class-balanced reweighting.

2. Unfair Experimental Comparison. EC2Face is implemented on the large FLUX-dev model, while most baselines (TediGAN, PixelFace+, MM2Latent, etc.) are significantly smaller GAN or Stable Diffusion models. This leads to unfair comparisons and likely exaggerates performance improvements. A fair study should include size-matched diffusion baselines or report results normalized by model capacity.
It remains unclear how the method behaves when applied to other backbones (e.g., Stable Diffusion 2.1 or SDXL).

3. There is no ablation study analyzing the real contribution of Temporal Dynamic Modulation, making its effectiveness unsubstantiated.

**Questions:**

See weaknesses

---

### Official Review · Reviewer_2Mra · 2025-11-11

**Soundness:** 2
**Presentation:** 3
**Contribution:** 2
**Rating:** 4
**Confidence:** 3

**Summary:**

This paper proposes EC2Face, a multimodal face generator built on the DiT backbone. There are two main additions to the model: (1) the explicit conditional consistency guidance that decodes a reverse-estimated image during training and enforces the CLIP similarity in the latent space and the mask-image consistency in pixel space. (2) A long-tail adaptive learning scheme that reweights the flow-matching loss to emphasize rare mask attributes. The method reports higher mask and text consistency in the tested datasets.

**Strengths:**

1. This paper addresses a real gap in face generation, namely, the insufficient fine-grained alignment to the different conditions. The method is simple and compatible with the existing architecture.

2. The results on the standard dataset MM-CElebA-HQ show competitive results compared to existing baseline methods. The gradient reweighting method shows quite significant gains on rare attributes, which resolves one of the challenges that the authors are claiming.

**Weaknesses:**

1. The method uses FaRL as the face parser inside the training loss and evaluates the mask accuracy with a parser as well. The author did not discuss whether a different parser is used or not. The authors should test with different independent parsers to avoid the coupling.

2. The baselines include mostly GAN or early diffusion control methods. There is no comparison with recent DiTs or rectified flow-based methods.

3. The ablation study doesn't isolate all the design choices. For example, there is no ablation on the temporal modulation hyperparameters, and where and when to decode $\tilde{x}$.

4. The identity and realism metrics are not sufficient, e.g., the face recognition similarity could be added as a metric, or a user study for semantic faithfulness and visual quality.

5. The contributions feel to be incremental relative to the current DiT-based method, and the application scope is narrow.

**Questions:**

1. In the reverse estimation step, the proposed equation (5) seems to be valid only under a linear coupling assumption, a short proof that this is consistent with the flow-matching path, while stating the assumptions will be helpful.

2. The logistic gate in equation (8) is reasonable, but the choices of the exact parameters are not explained. The authors could consider providing a quantitative reliability curve to showcase the reason for the choices.

3. The assumption that reverse estimates are not reliable for larger t should be demonstrated.

4. The weight in the equation (10) increases emphasis on rare labels, but will this cause a stability issue when there is a very large weight?
Will this over-penalize the rare-class pixels? Probably a more detailed discussion on this matter would be helpful.

---

### Note · Authors · 2026-01-26

I have read and agree with the venue's withdrawal policy on behalf of myself and my co-authors.

---

### Meta-Review · Area_Chair_XCom · 2025-12-14

**Summary:**

The reviewers' concerns are mainly about 1) novelty, 2) comparisons, 3) methodology, and 4) evaluation. For example, the comparison is unfair since the proposed model, which is based on FLUX, is significantly larger than the methods in comparison (e.g., GANs). Also, reviewers also point out that ablation is insufficient to show the effectiveness of the models.

**Reviewer Concerns:**

There is no rebuttal provided and the AC agrees with the reviewers' concerns.

**Reviewer Scores:**

This paper receives an initial rating of (2, 4, 4, 4). There is no rebuttal and response. Therefore, the AC believes that the final rating will remain unchanged. After reading the paper and review. The AC recommends a rejection.

---

### Decision · Program_Chairs · 2026-01-26

Reject